# Activin A Promotes Osteoblastic Differentiation of Human Preosteoblasts through the ALK1-Smad1/5/9 Pathway

**DOI:** 10.3390/ijms222413491

**Published:** 2021-12-16

**Authors:** Hideki Sugii, Mhd Safwan Albougha, Orie Adachi, Hiroka Tomita, Atsushi Tomokiyo, Sayuri Hamano, Daigaku Hasegawa, Shinichiro Yoshida, Tomohiro Itoyama, Hidefumi Maeda

**Affiliations:** 1 Department of Endodontology and Operative Dentistry, Division of Oral Rehabilitation, Faculty of Dental Science, Kyushu University, Fukuoka 812-8582, Japan; mhdsafwanbouga@gmail.com (M.S.A.); orie@dent.kyushu-u.ac.jp (O.A.); hiroka048@dent.kyushu-u.ac.jp (H.T.); shamano@dent.kyushu-u.ac.jp (S.H.); hide@dent.kyushu-u.ac.jp (H.M.); 2 Department of Endodontology, Kyushu University Hospital, Fukuoka 812-8582, Japan; tomokiyo@dent.kyushu-u.ac.jp (A.T.); daigaku8@dent.kyushu-u.ac.jp (D.H.); s.yosida@dent.kyushu-u.ac.jp (S.Y.); itoyama@dent.kyushu-u.ac.jp (T.I.); 3 OBT Center, Faculty of Dental Science, Kyushu University, Fukuoka 812-8582, Japan

**Keywords:** activin A, preosteoblast, osteoblastic differentiation, periodontal ligament, ALK1

## Abstract

Activin A, a member of transforming growth factor-β superfamily, is involved in the regulation of cellular differentiation and promotes tissue healing. Previously, we reported that expression of activin A was upregulated around the damaged periodontal tissue including periodontal ligament (PDL) tissue and alveolar bone, and activin A promoted PDL-related gene expression of human PDL cells (HPDLCs). However, little is known about the biological function of activin A in alveolar bone. Thus, this study analyzed activin A-induced biological functions in preosteoblasts (Saos2 cells). Activin A promoted osteoblastic differentiation of Saos2 cells. Activin receptor-like kinase (ALK) 1, an activin type I receptor, was more strongly expressed in Saos2 cells than in HPDLCs, and knockdown of ALK1 inhibited activin A-induced osteoblastic differentiation of Saos2 cells. Expression of ALK1 was upregulated in alveolar bone around damaged periodontal tissue when compared with a nondamaged site. Furthermore, activin A promoted phosphorylation of Smad1/5/9 during osteoblastic differentiation of Saos2 cells and knockdown of ALK1 inhibited activin A-induced phosphorylation of Smad1/5/9 in Saos2 cells. Collectively, these findings suggest that activin A promotes osteoblastic differentiation of preosteoblasts through the ALK1-Smad1/5/9 pathway and could be used as a therapeutic product for the healing of alveolar bone as well as PDL tissue.

## 1. Introduction

Periodontal tissue includes cementum, gingiva, periodontal ligament (PDL), and alveolar bone [1,2]. The primary functions of periodontal tissue are to support the teeth, help to attach the tooth to the surrounding tissues, and provide sensory responses to occlusal force [3,4]. However, once periodontal tissue has been irreversibly damaged by trauma, severe periodontitis, or deep caries, it is difficult to regenerate these tissues. Therefore, various studies have analyzed potential methods for enhancing the regeneration of periodontal tissue.

The regeneration of periodontal tissue is regulated by the local production of growth factors. One growth factor, activin A, is a member of the transforming growth factor-β (TGFβ) superfamily [5,6], and is a homodimer of two βa subunits of inhibin. The gene of activin A is called inhibinβa [7]. This factor is localized in many tissues, including the bone marrow, central nervous system, heart, liver, ovaries, placenta, and spleen [8,9,10,11], and has been reported to be related to the regulation of cellular proliferation [12,13], differentiation [14,15], and migration [16,17] in these tissues.

The TGFβ superfamily molecules bind and signal through the receptor complexes that consist of a dimer of activin type I receptors and a dimer of activin type II receptors [18,19]. After TGFβ superfamily molecules bind the receptor complexes, activin type II receptors activate the activin type I receptors, and then activin type I receptors activate intracellular signaling through the phosphorylation of Smads, which are the major signaling effectors of TGFβ superfamily molecules [20,21]. 

The TGFβ superfamily has two subfamilies: the TGFβ/nodal/activin subfamily and the bone morphogenetic protein (BMP)/growth differentiation factor (GDF) subfamily. In the TGFβ/nodal/activin subfamily, activin receptor-like kinase (ALK) 4 and ALK5, which are activin type I receptors, are related to activation of the phosphorylation of Smad2/3. Other activin type I receptors, such as ALK1, ALK2, ALK3 and ALK6, are related to activation of the phosphorylation of Smad1/5/9 in the BMP/GDF subfamily [22]. 

Activin A is a member of the TGFβ/nodal/activin subfamily, and activin A-induced cellular activities are mainly activated by the ALK4-Smad2/3 pathway [23,24,25]. However, some reports claim that activin A also regulates the signaling pathway of the BMP/GDF subfamily [26,27]. Among the activin type I receptors of the BMP/GDF subfamily, activin A is reported to bind to ALK1, ALK2 and ALK6 [28]. Furthermore, previous reports have demonstrated that the ALK1-mediated pathway induces osteoblastic differentiation of myoblasts [29,30] and mesenchymal stem cells [31,32] through BMP9 signaling. However, there are no reports about the effects of activin A on ALK1-mediated induction of osteoblastic differentiation.

Previous studies have reported that expression of activin A was detected in mouse dental follicles, which contains the progenitors of PDL cells and osteoblasts [33,34,35]. We previously focused on the effects of activin A on human PDL cells (HPDLCs) using in vitro study and rat PDL tissue using in vivo study. We reported that activin A promoted the proliferation, migration, and PDL-related gene expression of HPDLCs, and was expressed in PDL tissue, and its expression was increased around damaged periodontal tissue along with expression of interleukin-1β (IL-1β), which was coexpressed during the healing process in rats [36]. We also observed activin A expression on the surface of alveolar bone around damaged periodontal tissue [36]. However, the biological functions of activin A in preosteoblasts, which are distributed in alveolar bone, have not been reported.

This study was thus performed to evaluate the effect of activin A on preosteoblasts (Saos2 cells) by specifically analyzing (1) the expression of activin A in Saos2 cells, (2) the effects of activin A on the osteoblastic differentiation of Saos2 cells, and (3) the signaling pathway of activin A-induced osteoblastic differentiation of Saos2 cells.

## 2. Results

### 2.1. Expression of Activin A in Rat PDL Tissue and Saos2 Cells

Immunohistochemical analysis resulted in a positive reaction with an anti-activin A antibody in normal rat PDL tissue, with an especially intense reaction in the osteoblastic layer of the alveolar bone (Figure 1A). Immunocytochemical analysis also exhibited a positive reaction with an anti-activin A antibody in Saos2 cells (Figure 1B). Control staining (in the application of the control IgG) showed a negative reaction in PDL tissues and Saos2 cells (Figure 1A,B). In addition, immunofluorescent analysis revealed that expressions of activin A and osteocalcin (OCN) were colocalized in the osteoblastic layer of the alveolar bone in rat periodontal tissue (Figure 1C).

### 2.2. Effects of Activin A on Osteoblastic Differentiation of Saos2 Cells

Saos2 cells were cultured with or without activin A (5, 10, 50, or 100 ng/mL) in the presence of 0.5 mM CaCl_2_ to analyze the effect of activin A on osteoblastic differentiation of preosteoblasts. Alizarin Red S staining (Figure 2A) and von Kossa staining (Figure 2B) both demonstrated a dose-dependent promotion of mineralization in Saos2 cells treated with activin A. Furthermore, expression of *OCN*, *osteopontin* (*OPN),* and *runt-related transcription factor 2 (RUNX2)* were upregulated in activin A-treated Saos2 cells (Figure 2C). We also analyzed alkaline phosphatase (ALP) activity and gene expression of *ALP* in Saos2 cells treated with or without CaCl_2_ and activin A. ALP activity and *ALP* expression were upregulated in activin A-treated Saos2 cells with CaCl_2_ compared with Saos2 cells with CaCl_2._ On the other hand, Saos2 cells without CaCl_2_ exhibited no promotion effects of ALP activity and *ALP* expression, and activin A-treated Saos2 cells without CaCl_2_ also showed no effects on ALP activity and *ALP* expression (Appendix A). These results indicate that activin A promoted osteoblastic differentiation of Saos2 cells.

### 2.3. Effects of Activin Type I Receptors on Activin A-Induced Osteoblastic Differentiation in Saos2 Cells

For analyzing the biological mechanism of activin A-induced osteoblastic differentiation in Saos2 cells, we focused on the activin type I receptors, ALK1-6. First, we analyzed ALK4 and ALK5, which are the receptors of the TGFβ/nodal/activin subfamily. We examined the effects of siALK4 or siALK5 on activin A-induced osteoblastic differentiation of Saos2 cells. Neither siALK4 nor siALK5 had any effect on activin A-induced mineralization and bone-related gene (*OCN*, *OPN,* and *RUNX2*) expression in Saos2 cells (Appendix A).

Next, to evaluate the contribution of ALK1, which is the receptor of the BMP/GDF subfamily, on the osteoblastic differentiation of Saos2 cells, we examined the effect of siALK1 on activin A-induced osteoblastic differentiation in Saos2 cells. Expression of *ALK1* in Saos2 cells was downregulated by siALK1 (Figure 3A) and siALK1 had no effects on the proliferation of Saos2 cells (Figure 3B). Saos2 cells treated with siALK1 were cultured with or without 100 ng/mL activin A in the presence of 0.5 mM CaCl_2_. The treatment with siALK1 inhibited activin A-induced mineralization of Saos2 cells compared with control siRNA (Figure 3C). In addition, the expression levels of *OCN*, *OPN,* and *RUNX2* were significantly lower in Saos2 cells treated with siALK1 than in cells treated with control siRNA in the presence of activin A (Figure 3D). We confirmed the same effects of siALK1 on activin A-induced osteoblastic differentiation in NOS1 cells, which are another source of preosteoblasts (Appendix A).

We also analyzed the effect of ALK2, ALK3 and ALK6, which are also receptors of the BMP/GDF subfamily, on activin A-induced osteoblastic differentiation of Saos2 cells. The treatment with siALK2, siALK3 or siALK6 had no effect on activin A-induced minealization and bone-related gene (*OCN*, *OPN,* and *RUNX2*) expression in Saos2 cells (Appendix A).

### 2.4. Expression of ALK1 in Rat PDL Tissue

The results of immunohistochemical analysis showed a positive reaction of an anti-ALK1 antibody in the PDL tissue, and the osteoblastic layer of the alveolar bone was intensely stained (Figure 4A). Western blotting analysis also revealed that expression of ALK1 was significantly higher in Saos2 cells than in HPDLCs, consistent with the immunohistochemical results (Figure 4B).

### 2.5. Expression of ALK1 in Surgically Damaged PDL Tissue

We compared the immunolocalization of ALK1 between normal and damaged periodontal tissue in rats (Figure 5A,B). More ALK1-immunoreactive cells were detected in the osteoblastic layer of the alveolar bone around the damaged site (Figure 5C) compared with the tissue around the nondamaged second molar on the damaged site and the first molar on the control site (Figure 5D,E). Next, we investigated the effects of the proinflammatory cytokines, IL-1β and tumor necrosis factor-α (TNF-α), on the gene and protein expression of ALK1 in Saos2 cells using quantitative RT-PCR and Western blotting analysis, respectively (Figure 5F,G). Expression of ALK1 was more strongly upregulated by IL-1β and TNF-α in Saos2 cells than in untreated cells. In addition, the expression of *inhibin**βa*, which is the gene of activin A, was significantly upregulated after treatment with IL-1β and TNF-α in Saos2 cells in comparison with untreated cells (Figure 5H). We also analyzed the expression of *ALK1* in activin A-treated Saos2 cells and we found the expression of *ALK1* was upregulated by activin A treatment in Saos2 cells. In addition, after IL-1β or TNF-α treatment, expression of ALK1 was further upregulated in activin A-treated Saos2 cells (Appendix A).

### 2.6. Effects of ALK1 Knockdown on Phosphorylation of Activin A-Related Intracellular Signaling Molecules in Saos2 Cells

The activin A-related intracellular signaling molecules, Smad1/5/9 and Smad2/3, were analyzed during osteoblastic differentiation of Saos2 cells. Western blotting analysis demonstrated that phosphorylation of Smad1/5/9 was promoted by activin A during osteoblastic differentiation of Saos2 cells (Figure 6), whereas phosphorylation of Smad2/3 was not upregulated (Appendix A). In contrast, activin A-induced phosphorylation of Smad1/5/9 was inhibited by siALK1 compared with control siRNA (Figure 6A,B). The quantification of t-Smad1/5/9 expression exhibited no significant differences after activin A, CaCl_2_ and siALK1 treatment (Figure 6C). Furthermore, phosphorylation of Smad2/3 was not changed by siALK1 in Saos2 cells (Appendix A). These results suggest that activin A promotes osteoblastic differentiation of Saos2 cells through the ALK1-Smad1/5/9 pathway.

## 3. Discussion

This study demonstrates the effect of activin A on osteoblastic differentiation of Saos2 cells and reveals the activin A-induced signaling pathway. Activin A promoted osteoblastic differentiation of Saos2 cells and could activate these signals via the ALK1-Smad1/5/9 pathway. Here, we show for the first time that activin A is important for activating the ALK1 (which is the receptor of the BMP/GDF subfamily)-mediated osteoblastic differentiation of human preosteoblasts.

Activin A is expressed in bone marrow cells and is related to bone metabolism [37,38]. Previous studies demonstrated that application of activin A in the parietal bone led to upregulation of bone formation in newborn rats [39] and mice [40]. An in vitro study demonstrated that activin A contributed to osteogenesis of mouse embryonic stem cells [41] and promoted osteoblastic differentiation of murine bone marrow cells [38] and murine myoblasts [42]. Consistent with these reports, our results demonstrated that activin A was expressed in Saos2 cells and promoted osteoblastic differentiation of Saos2 cells. Furthermore, activin A promoted ALP activity in Saos2 cells treated with calcium. The past report showed that activin A increased the intracellular Ca^2+^ concentration [43]. These results suggest that activin A promoted osteoblastic differentiation of Saos2 cells through increasing the intracellular Ca^2+^ concentration.

However, several studies have revealed that activin A inhibited the osteoblastic differentiation of rat calvarial cells [44,45], ALP activity by inhibition of extracellular matrix mineralization in human bone marrow-derived mesenchymal stem cells [46] and we previously reported that activin A inhibited osteoblastic differentiation of HPDLCs [36]. These opposing effects of activin A on osteoblastic differentiation may be caused by the basis of experimental conditions, such as cell type and differentiation status [7,11,47]. Thus, further studies elucidating the biological mechanism of activin A-induced osteoblastic differentiation are required.

In this study, we used human osteosarcoma cells, Saos2 cells and NOS1 cells, as preosteoblasts. Previous study revealed that activin A promoted the proliferation, invasion, and migration of human osteosarcoma cells [48]. However, there are no reports about the effects of activin A on the differentiation of human osteosarcoma cells. Our results showed for the first time that activin A promoted the osteoblastic differentiation of human osteosarcoma cells, Saos2 cells and NOS1 cells.

Activin A usually signals via the ALK4-Smad2/3 pathway, and previous reports showed that the ALK4-mediated pathway was related to fibroblastic differentiation of skin fibroblasts [49] and atrial fibroblasts [50]. However, our present results indicate that ALK4 and ALK5, which are the receptors of the TGFβ/nodal/activin subfamily, did not affect activin A-induced osteoblastic differentiation of Saos2 cells (Appendix A), suggesting that ALK4 and ALK5 are not the key receptors for the activin A-induced signaling pathway during osteoblastic differentiation of Saos2 cells.

Some studies reported that activin A can also regulate the signaling pathway of the BMP/GDF subfamily [26,27]. It was reported that activin A could activate phosphorylation of Smad1/5/9 and that treatment with the inhibitors of ALK1, ALK2, ALK3 and ALK6, which are the receptors of the BMP/GDF subfamily, inhibited activin A-induced phosphorylation of Smad1/5/9 [26]. In this study, we showed that siALK1 led to inhibition of activin A-induced osteoblastic differentiation and phosphorylation of Smad1/5/9, whereas siALK2, siALK3, and siALK6 had no effect on activin A-induced osteoblastic differentiation in Saos2 cells. ALK1 is related to osteoblastic differentiation in various types of cells, such as endothelial cells [51], chondrocytes [52,53], myoblasts [29,30], and mesenchymal stem cells [31,32]. These results suggest that the ALK1-Smad1/5/9 pathway is important for activin A-induced osteoblastic differentiation of Saos2 cells. 

Damaged periodontal tissue requires the processes of inflammation, proliferation, and tissue remodeling for healing to occur [54,55,56]. During the inflammatory process, the expression of the proinflammatory cytokines, IL-1β and TNF-α, is accelerated. In addition, expression of inhibinβa was shown to be upregulated in osteoprogenitor cells during the inflammatory process after surgery of a rat femur [57], and our previous study also showed that expression of activin A was increased in PDL tissue and alveolar bone around damaged periodontal tissue, with increased expression of IL-1β [36]. In this study, we showed that the expression of inhibinβa and ALK1 was upregulated in IL-1β- or TNF-α-treated Saos2 cells. We also exhibited that expression of ALK1 was promoted by activin A in Saos2 cells, and the treatment with IL-1β or TNF-α revealed further upregulation of ALK1 expression in activin A-treated Saos2 cells. Furthermore, expression of ALK1 was higher in the osteoblastic layer of alveolar bone around the damaged periodontal tissue compared with normal tissue. Previous studies reported that expression of ALK1 was upregulated after neuronal injury in the brain, and that ALK1 was related to the healing of the injured neurons [58]. These results suggest that activin A acts on the autocrine or paracrine manner in preosteoblasts, and activin A and ALK1 play an important role in the healing of periodontal tissue including alveolar bone, especially during the inflammatory process. 

Our previous study demonstrated that expression of ALK4 was higher in HPDLCs than in Saos2 cells, and that activin A could activate the ALK4-Smad2/3 pathway during upregulation of PDL-related gene expression in HPDLCs (data not shown). However, our present study revealed that expression of ALK1 was higher in Saos2 cells than in HPDLCs and that activin A promoted osteoblastic differentiation of Saos2 cells through the ALK1-Smad1/5/9 pathway. Our findings suggest that the expression pattern of activin type I receptors differs according to the type of cell, and activin A has bifunctional effects on the healing of damaged periodontal tissue through activation of the different signaling pathways in PDL cells and preosteoblasts. Activin A acts firstly on PDL cells by the upregulation of PDL-related gene expression through the ALK4-Smad2/3 pathway, and secondly on preosteoblasts by the promotion of osteoblastic differentiation through the ALK1-Smad1/5/9 pathway (Figure 7). Thus, activin A could be used as an effective therapeutic product for healing damaged periodontal tissue, including PDL tissue and alveolar bone, by applying its bifunctional effects.

## 4. Materials and Methods

### 4.1. Cell Culture

Saos2 cells (RIKEN, Saitama, Japan) and NOS1 cells (RIKEN) were used as human preosteoblasts. We isolated HPDLCs from the extracted third molar of a 23-year-old man who was a healthy patient at Kyushu University Hospital. The cells were used from passage 4 through 7, as described previously [36]. The medium used for cell culture consisted of alpha-minimum essential medium (αMEM; Gibco-BRL, Grand Island, NY, USA) supplemented with 10% fetal bovine serum (FBS; Biosera, Nuaillé, France), 50 µg/mL streptomycin, and 50 U/mL penicillin (10%FBS/αMEM). The cells were cultured at 37 °C in a humidified atmosphere of 5% CO_2_ and 95% air. All procedures for HPDLCs were undertaken in compliance with the Research Ethics Committee, Faculty of Dentistry, Kyushu University. 

### 4.2. Animal Model

This study used 5-week-old male Sprague Dawley rats (Kyudo, Saga, Japan), weighing 140–150 g (n = 4). All procedures were undertaken in accordance with our previous report [59]. Anesthesia, which included 0.15 mg/kg medetomidine (Kyoritsu Seiyaku Co. Ltd., Tokyo, Japan), 2.5 mg/kg butorphanol tartrate (Meiji Seika Pharma Co. Ltd., Tokyo, Japan), and 2 mg/kg midazolam (Sandoz Inc, Tokyo, Japan), was injected intraperitoneally. A periodontal defect was created in the left maxillary bone extending from the mesiopalatal submarginal portion corresponding to the distopalatal portion of the first molar (2 mm in diameter and 2 mm in depth). The right maxillary bone was not damaged and was used as a control. All procedures were approved by the Animal Ethics Committee and conformed with the regulations of Kyushu University (approval code: A30-265-0). At 3 days after creation of the periodontal defect, transcardial perfusion was performed using 4% paraformaldehyde (Merck, Darmstadt, Germany) in phosphate-buffered saline (PBS). Samples were collected from the maxillae and decalcified in 10% ethylenediaminetetraacetic acid (Wako Pure Chemical Industries Ltd., Osaka, Japan) at 4 °C for 1 month. After dehydration, OCT compound (Sakura Finetek, Tokyo, Japan) was used for embedding the samples.

### 4.3. Western Blotting Analysis 

Saos2 cells were cultured until subconfluency and then cultured under serum starvation for 2 h. The cells were treated with or without 100 ng/mL activin A (PeproTech EC, London, UK) in the presence of calcium chloride (CaCl_2_; 0.5 mM) for 30 min and lysed in Pierce radioimmunoprecipitation assay buffer (Invitrogen, Waltham, MA, USA). Aliquots containing 20 µg protein per lane were prepared for 10% sodium dodecylsulfate polyacrylamide gel electrophoresis and then transferred onto Immune-Blot polyvinylidene difluoride (PVDF) membrane (Bio-Rad Laboratories, Hercules, CA, USA). The membrane was reacted with the following antibodies, mouse monoclonal anti-β-actin (Santa Cruz, Dallas, TX, USA) at a dilution of 1:1000, rabbit polyclonal anti-ALK1 (Abcam, Cambridge, UK) at a dilution of 1:500, rabbit polyclonal anti-phospho Smad2/3 (Cell Signaling, Beverly, MA, USA) at a dilution of 1:1000, rabbit polyclonal anti-total Smad2/3 (Cell Signaling) at a dilution of 1:1000, rabbit polyclonal anti-phospho Smad1/5/9 (Cell Signaling) at a dilution of 1:1000, or rabbit polyclonal anti-total Smad1/5/9 (Santa Cruz) at a dilution of 1:1000. The biotinylated anti-mouse IgG (Nichirei Biosciences Inc., Tokyo, Japan) or anti-rabbit IgG (Nichirei Biosciences) was added to the reacted membrane, and then the results were evaluated by an ECL Select Western blotting detection system (GE Healthcare, Buckinghamshire, UK). BLUeye Prestained Protein Ladder (Gene DireX Inc., Las Vegas, NV, USA) was selected as the standard protein.

### 4.4. Immunofluorescence and Immunochemical Analyses

For immunofluorescence analysis, tissue sections were prepared (5 µm) and then immersed with the blocking agent using 2% bovine serum albumin (BSA) in PBS for 1 h. The sections were reacted with a goat polyclonal anti-activin A antibody (R&D systems, Minneapolis, MN, USA) at a concentration of 5 µg/mL, a rabbit polyclonal anti-ALK1 antibody (Abcam) at a dilution of 1:100, or a rabbit polyclonal anti-OCN antibody at a dilution of 1:250 overnight at 4 °C, followed by Alexa-Fluor488-conjugated anti-rabbit IgG (Invitrogen) or Alexa-Fluor647-conjugated anti-goat IgG (Invitrogen) secondary antibodies for 1 h. VECTASHILD Mounting Medium containing 4,6-diamidino-2-phenylindole (DAPI; Vector Laboratories, Burlingame, CA, USA) was used for the staining of nuclei. Imaging of the sections was performed using a Biozero digital microscope (Keyence Corporation, Osaka, Japan).

For immunochemical analysis, tissue sections were immersed with the blocking agent using 2% BSA in PBS for 1 h. The sections were reacted for 1 h with a goat polyclonal anti-activin A antibody (R&D systems) at a concentration of 5 µg/mL, and a rabbit polyclonal anti-ALK1 antibody (Abcam) at a dilution of 1:100. The sections were then reacted with a biotinylated secondary antibody (Nichirei Biosciences), followed by avidin-peroxidase conjugate (Nichirei Biosciences) and DAB solution (Nichirei Biosciences). Mayer’s hematoxylin solution (Wako Pure Chemical Industries Ltd.) was used for the staining of nuclei.

### 4.5. Treatment of Saos2 Cells with IL-1β or TNF-α

Treatment of Saos2 cells (6 × 10^4^ cells per well) with 10 ng/mL recombinant human IL-1β (PeproTech EC) or 10 ng/mL recombinant human TNF-α (PeproTech EC) was examined in 2 mL of culture medium for 1 day. The concentration of these cytokines was determined based on the methods used in a previous study [60].

### 4.6. Quantitative Reverse Transcription Polymerase Chain Reaction (RT-PCR)

An ExScript RT reagent kit (Takara Bio, Shiga, Japan) was used for synthesis of first-strand cDNA from 1 mg total RNA. Reverse transcription of total RNA was performed with random 6-mers and ExScriptRTase for 15 min at 42 °C, followed by incubation for 2 min at 99 °C and 5 min at 5 °C to stop the reaction. A KAPA SYBR FAST qPCR Kit (Kapa Biosystems Inc., Boston, MA, USA) was used for PCR under the following conditions: 95 °C for 10s (initial denaturation), then 40 cycles of 95 °C for 5s and 60 °C for 30s, followed by a dissociation program of 95 °C for 15s, 60 °C for 30s, and 95 °C for 15s. We used human *β-actin* as an internal control and the 2^(−ΔΔCt)^ values for calculation of the target gene expression. Primer sequences, annealing temperatures, and product sizes for ALK1, ALK2, ALK3, ALK4, ALK5, ALK6, ALP, OCN, OPN, RUNX2 and β-actin are shown in Table 1. The GenBank database (NCBI) was used for designing primer sequences, and specificity of the primer sequences was ensured by performing a BLAST search of GenBank. Normalization of the target genes was performed against β-actin expression, and the values were exhibited as the fold increase of the control.

### 4.7. Osteoblastic Differentiation of Saos2 Cells

Saos2 cells (1 × 10^4^ cells per well) were cultured in 24-well plates with or without activin A (5, 10, 50 or 100 ng/mL) in the presence of 0.5 mM CaCl_2_ for 3 weeks. The cells were stained using Alizarin red S staining and von Kossa staining. The Alizarin red-positive areas and the von Kossa-positive areas were quantified by a Biozero digital microscope (Keyence Corporation). Total RNA was isolated after 2 weeks of culture under the same conditions, and the bone-related gene expression was analyzed.

### 4.8. Small Interfering RNA Transfection

Small interfering RNAs (siRNAs) for ALK1 (siALK1, MISSION siRNA; Sigma-Aldrich, St. Louis, MO, USA), ALK2 (siALK2, MISSION siRNA; Sigma-Aldrich), ALK3 (siALK3, MISSION siRNA; Sigma-Aldrich), ALK4 (siALK4, MISSION siRNA; Sigma-Aldrich), ALK5 (siALK5, MISSION siRNA; Sigma-Aldrich), ALK6 (siALK6, MISSION siRNA; Sigma-Aldrich), and nontargeting control (siControl, MISSION siRNA Universal Negative Control; Sigma-Aldrich) were used for this study. One day before transfection, Saos2 cells were cultured in Opti-MEM I medium on 24-well plates (Invitrogen) supplemented with 10% FBS (10%FBS/Opti-MEM). After the cells increased to 30–50% confluency, the Saos2 cells were transfected with siRNA. Each siRNA (20 pmol) in Opti-MEM I medium (50 µL) was mixed with Lipofectamine iMAX (1 µL; Invitrogen) in another Opti-MEM I medium (50 µL). The mixture was incubated for 5 min at room temperature, then mixed with the cell culture medium (500 µL 10% FBS/Opti-MEM) at 37 °C in a humidified atmosphere of 5% CO_2_ and 95% air. After 48 h of culture, the medium was replaced with fresh 10% FBS/α-MEM.

### 4.9. Proliferation Assay 

We used a WST-1 Cell Proliferation Assay Kit (Millipore Corp., Billerica, MA, USA). After Saos2 cells were transfected with siRNA, proliferation of the cells was measured on days 0, 1, 2, 3, 5 and 7 of culture. The WST-1 reagent was mixed with the culture medium. After 30 min, the optical density of the mixture was measured using an iMark microplate reader (Bio-Rad Laboratories, Hercules, CA, USA) at an absorbance of 450 nm. 

### 4.10. ALP Activity Assay

Saos2 cells were cultured with or without 100 ng/mL activin A and 0.5 mM CaCl_2_ for 1, 3, and 5 days. The cells were lysed, and ALP activity was analyzed using ALP assay kit (Takara Bio) according to the manufacture’s instruction.

### 4.11. Statistical Analysis

All values are expressed as means ± standard deviation (SD). Statistical analysis was performed using one-way ANOVA followed by the Benjamini–Hochberg method. The statistical significance was determined as a probability value of *p* < 0.05.

## Figures and Tables

**Figure 1 ijms-22-13491-f001:**
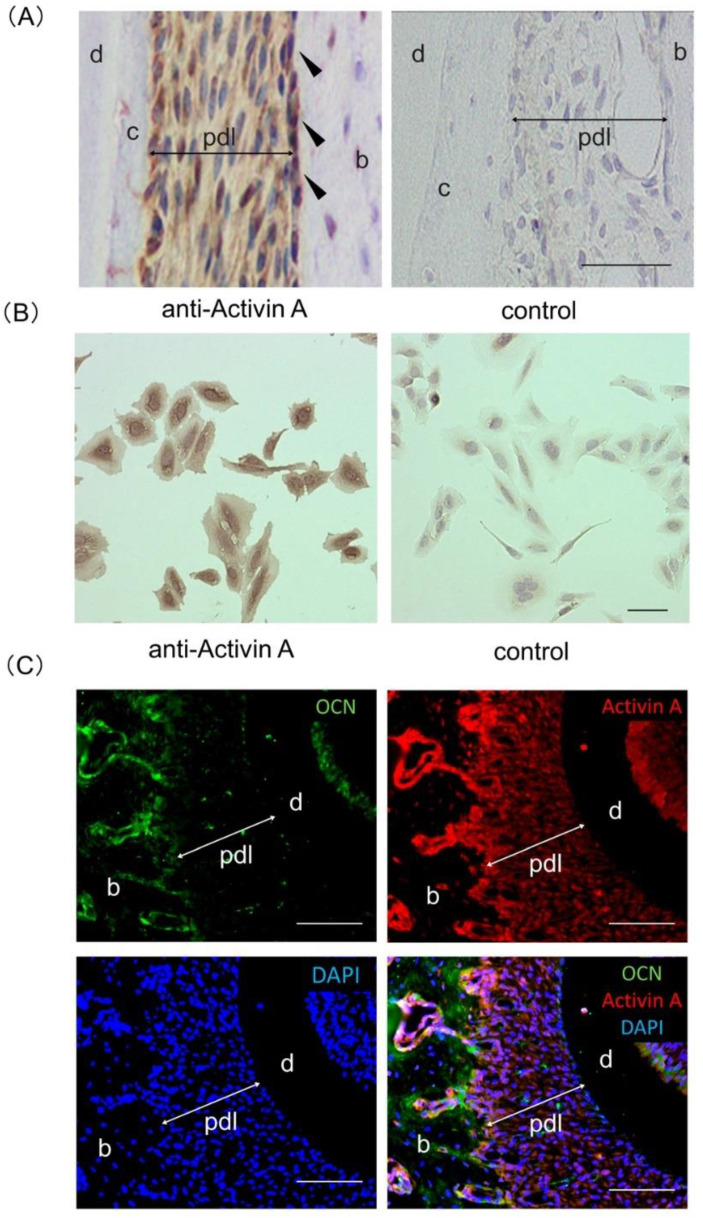
Expression of activin A in rat periodontal ligament (PDL) tissue and Saos2 cells. (**A**) Sagittal sections of rat PDL tissue were used for immunohistochemical analysis. A positive reaction with an anti-activin A antibody was detected in the PDL tissue and was particularly marked in the osteoblastic layer (black arrowhead). (**B**) Immunopositive staining against an anti-activin A antibody was found in the cytoplasm of Saos2 cells. Control IgG was used as a negative control. Hematoxylin was used for the staining of nuclei. Experiments were performed in duplicate. (**C**) Horizontal sections of rat PDL tissue were used for immunofluorescent analysis. Expression of activin A (red) and osteocalcin (OCN; green) were colocalized in osteoblastic layer of the alveolar bone (yellow). Nuclei were stained with 4,6-diamidino-2-phenylindole (DAPI; blue). b, alveolar bone; c, cementum; d, dentin; pdl, periodontal ligament. Bars = 100 µm.

**Figure 2 ijms-22-13491-f002:**
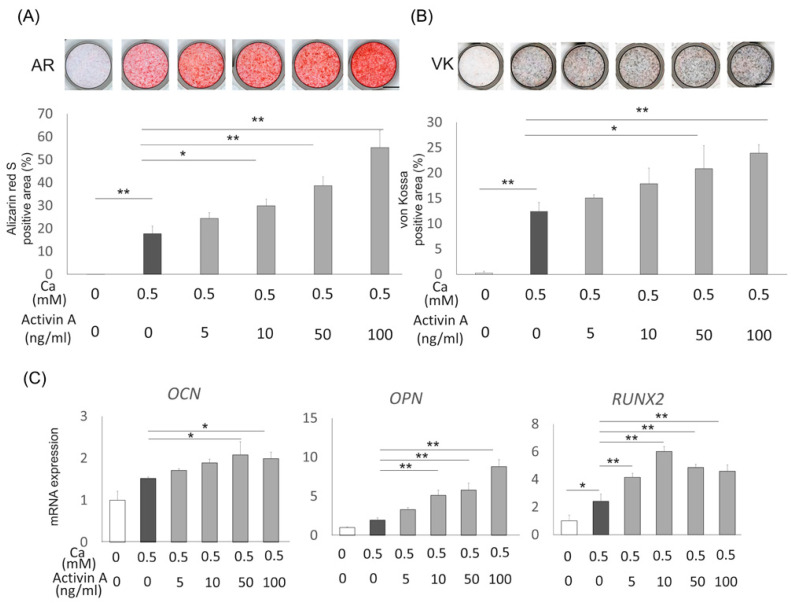
Effects of activin A on osteoblastic differentiation of Saos2 cells (**A**,**B**) Saos2 cells were cultured with 0.5 mM CaCl_2_ with or without activin A (5, 10, 50 or 100 ng/mL) for 3 weeks. Mineralization of Saos2 cells was evaluated by Alizarin Red S staining (AR) and von Kossa staining (VK), and quantification of the staining area was analyzed. (**C**) Expression of the genes encoding *OCN*, *OPN,* and *RUNX2* was analyzed using quantitative reverse transcription polymerase chain reaction (RT-PCR) in Saos2 cells cultured under the same conditions. Normalization of gene expression was performed against *β-actin* expression, and the results were evaluated as the fold increase of the control. Values are the means ± SD from three independent experiments. * *p* < 0.05, ** *p* < 0.01.

**Figure 3 ijms-22-13491-f003:**
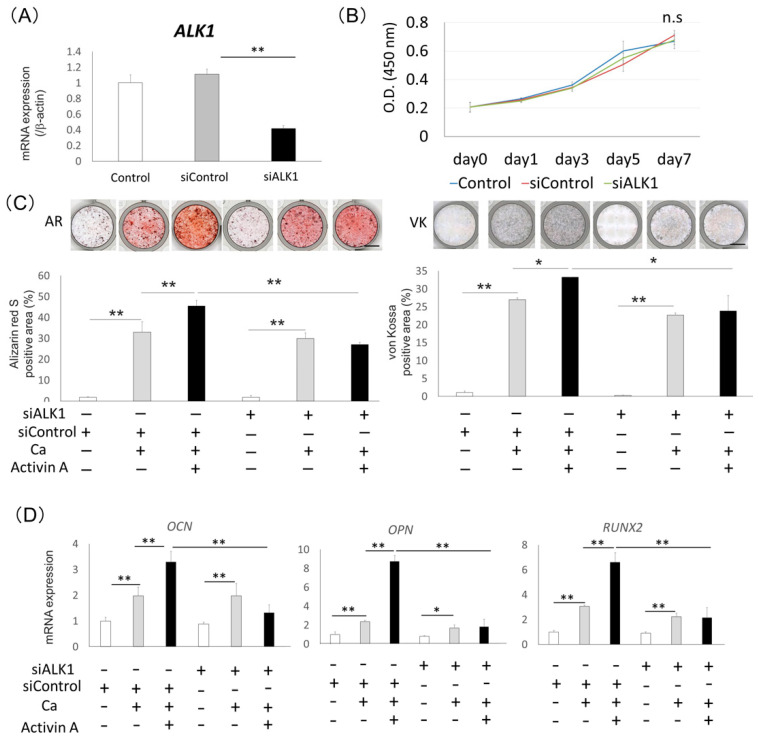
Treatment with siALK1 inhibited activin A-induced osteoblastic differentiation of Saos2 cells. (**A**) Gene expression of ALK1 in Saos2 cells with siALK1. Untreated cells were used as the control. Gene expression of ALK1 was analyzed using quantitative RT-PCR. Normalization of gene expression was performed against *β-actin* expression, and the gene expression levels were shown as the fold increase of the control. Values are the means ± SD from three independent experiments. ** *p* < 0.01. (**B**) Saos2 cells treated with siALK1 were cultured in 10%FBS/αMEM for 0, 1, 3, 5 and 7 days. A proliferation assay was performed using a WST-1 proliferation assay kit at an absorbance of 450 nm, n.s = no significance. (**C**) Saos2 cells were cultured with 0.5 mM CaCl_2_ (Ca) with or without 100 ng/mL activin A. Mineralization of Saos2 cells was evaluated by Alizarin Red S staining (AR) and von Kossa staining (VK), and quantification of the staining area was analyzed. * *p* < 0.05, ** *p* < 0.01. (**D**) Expression of bone-related genes (*OCN*, *OPN,* and *RUNX2*) in Saos2 cells treated with 100 ng/mL activin A was analyzed by quantitative RT-PCR. Untreated cells were used as a control. Normalization of gene expression was performed against *β-actin* expression, and the gene expression levels were shown as the fold increase of the control. * *p* < 0.05, ** *p* < 0.01.

**Figure 4 ijms-22-13491-f004:**
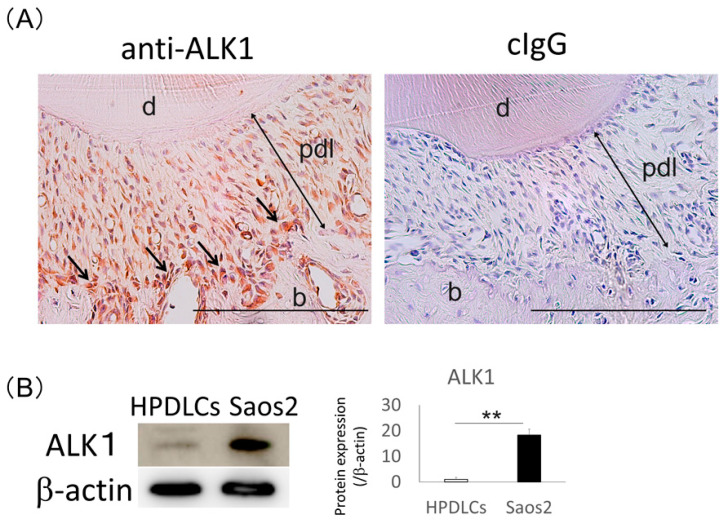
Expression of ALK1 in normal rat periodontal ligament (PDL) tissue and Saos2 cells. (**A**) Immunohistochemical analysis was performed using an anti-ALK1 antibody in rat PDL tissue. Positivity to an anti-ALK1 antibody was detected in PDL tissue, and ALK1-immunoreactive cells were localized in the osteoblastic layer (arrows). Control IgG was used for a negative control. Hematoxylin was used for the staining of nuclei. Experiments were performed in duplicate. b, alveolar bone; d, dentin; pdl, periodontal ligament. Bars = 500 µm. (**B**) Western blotting analysis was performed to detect expression of ALK1 in human PDL cells (HPDLCs) and Saos2 cells. The expression levels of this protein were normalized against β-actin expression and were quantified. The expression level of ALK1 was higher in Saos2 cells than in HPDLCs. Values are the means ± SD from three independent experiments. ** *p* < 0.01.

**Figure 5 ijms-22-13491-f005:**
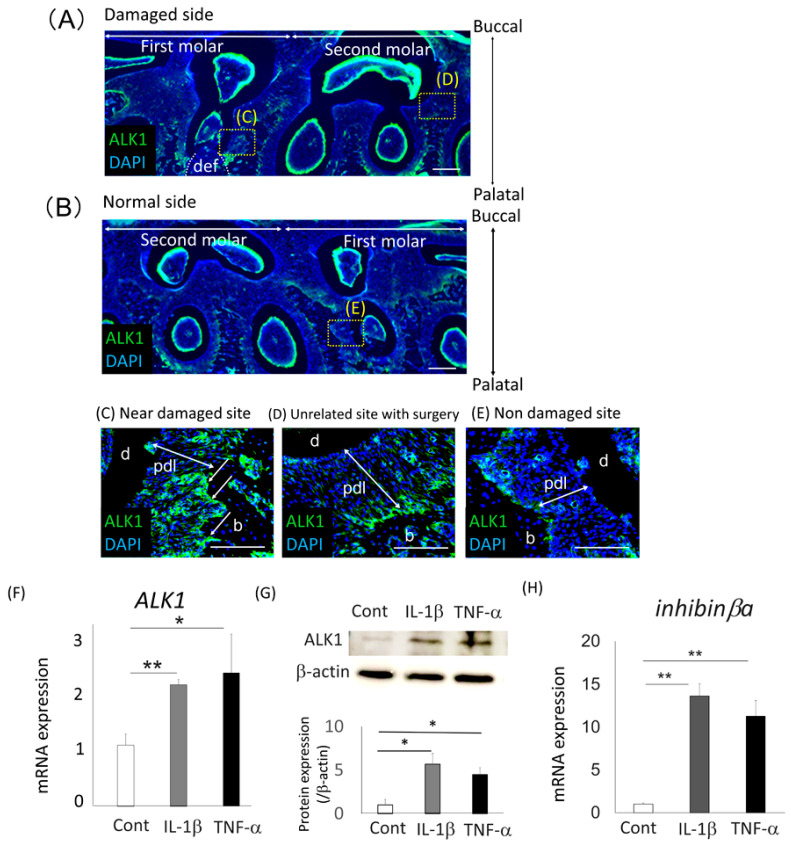
Expression of ALK1 around damaged rat periodontal tissue and in IL-1β- or TNF-α-treated Saos2 cells (**A**–**E**) Immunofluorescent analysis of ALK1 expression (green) in a surgically damaged side (**A**) and a normal side (**B**) of rat periodontal tissue was performed. The tissue samples were collected 3 days after surgery, and horizontal sections of the first and second molars were prepared from the rat maxilla. Panels C-E show higher magnification views of panels A and B. More intense staining was detectable around the osteoblastic layer in the defect site (def) (**C**), compared with normal periodontal tissue from the second molar in the damaged site (**D**) or the first molar from the normal site (**E**) (arrows). Nuclei were stained with 4,6-diamidino-2-phenylindole (DAPI; blue). Bars = 100 µm. b, alveolar bone; d, dentin; pdl, periodontal ligament. (**F**,**H**) Gene expression of *ALK1* (**F**) and inhibinβa (**H**) in Saos2 cells treated with IL-1β or TNF-α for 24 h was analyzed using quantitative RT-PCR. Untreated cells were used as the control. Normalization of gene expression was performed against *β-actin* expression, and the gene expression levels were shown as the fold increase of the control. Values were the means ± SD from three independent experiments. ** *p* < 0.01, * *p* < 0.05. (**G**) Western blotting analysis was performed to detect expression of ALK1 in Saos2 cells. Normalization of protein expression was performed against β-actin expression and the expression level of this protein was quantified. The results were shown as the fold increase of the control. Values are the means ± SD from three independent experiments. * *p* < 0.05.

**Figure 6 ijms-22-13491-f006:**
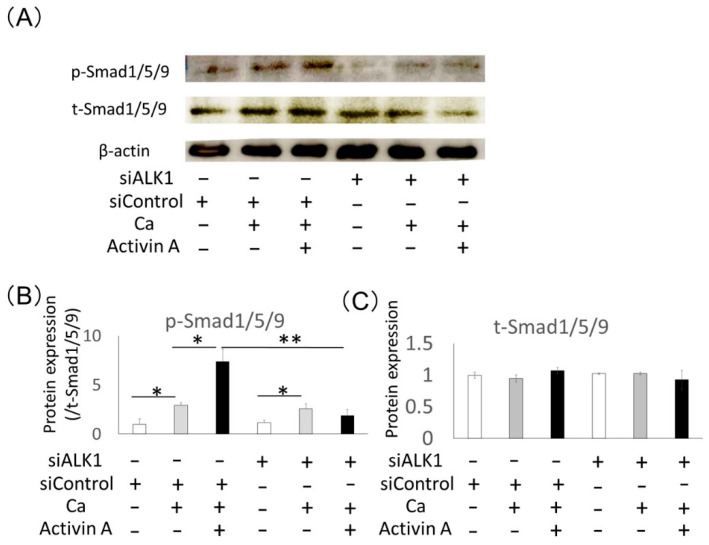
Treatment with siALK1 inhibited activin A-induced phosphorylation of Smad1/5/9 in Saos2 cells. (**A**) Protein expression of phospho-Smad1/5/9 (p-Smad1/5/9) and total-Smad1/5/9 (t-Smad1/5/9) was analyzed using Western blotting in Saos2 cells cultured with 0.5 mM CaCl_2_ (Ca) with or without 100 ng/mL activin A. β-actin was used as an internal standard for the control. (**B**,**C**) Expression levels of p-Smad1/5/9 were normalized against t-Smad1/5/9 expression, and the expression levels of t-Smad1/5/9 were quantified. The results are shown as the fold increase of the control. Values are the means ± SD from three independent experiments. * *p* < 0.05, ** *p* < 0.01.

**Figure 7 ijms-22-13491-f007:**
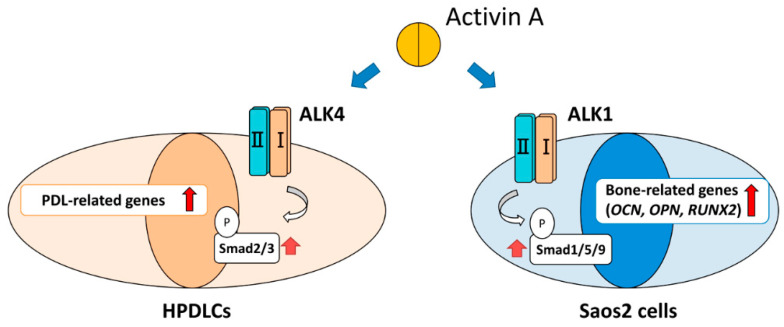
Schema of activin A-induced bifunctional effects on differentiation of HPDLCs and Saos2 cells.

**Table 1 ijms-22-13491-t001:** Primer sequence, product size, annealing temperature, for quantitative RT-PCR.

Target Gene (Abbreviation)	Forward (Top) and Reverse (Bottom) Primer Sequences	Size of Amplified Products (bp)	Annealing Temperature (°C)	Sequence ID
β-actin	5’-ATTGCCGACAGGATGCAGA-3’ 5’-GAGTACTTGCGCTCAGGAGGA-3’	89	60	NM_001101.3
ALK1	5’-ATCCTAGGCTTCATCGCCTC-3’ 5’-TAGCCTCAGAGCCAGATGGG-3’	141	60	NM_000020.3
ALK2	5’-AGGCTGCTTCCAGGTTTATGAG-3’ 5’-TGGCAGCACTCCACAGCTT-3’	81	60	NM_001105.5
ALK3	5’-AGTGTCTCCAGTCAAGCTCTGGGTA-3’ 5’-CCATCTCTGCTGCGCTCATTTA-3’	97	60	NM_004329.3
ALK4	5’-TGCTGCGCCATGAAAACATC-3’ 5’-GGGACCCGTGCTCATGATAG-3’	105	60	NM_020327.3
ALK5	5’-CCTTCTGACCCATCAGTTGAAGA-3’ 5’-CCTAGCTGCTCCATTGGCAT-3’	150	60	NM_004612.4
ALK6	5’-GTTACGCCCCTCATTCCCAA-3’ 5’-GTTTTCTTAACCCGCAGGGC-3’	126	60	NM_001256793.2
Inhibinβa	5’-TGTGCCCACCAAGCTGAGAC-3’ 5’-CTGGGCTGGGCAACTCTATGA-3’	124	60	NM_002192.2
OCN	5’-GTGCAGAGTCCAGCAAAGGT-3’ 5’-TCAGCCAACTCGTCACAGTC-3’	110	60	NM_199173.4
OPN	5’-ACACATATGATGGCCGAGGTGA-3’ 5’-TGTGAGGTGATGTCCTCGTCTGT-3’	115	60	NM_000582.3
RUNX2	5’-AACCCTTAATTTGCACTGGGTCA-3’ 5’-CAAATTCCAGCAATGTTTGTGCTAC-3’	145	60	NM_001024630.3

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
