# Peer review of "Activin A Promotes Osteoblastic Differentiation of Human Preosteoblasts through the ALK1-Smad1/5/9 Pathway"

_ijms, 2021, doi:10.3390/ijms222413491_

Round 1

Reviewer 1 Report

Review of International Journal of Molecular Sciences

Titled: Activin A promotes osteoblastic differentiation of human preosteoblasts through the ALK1-Smad1/5/9 pathway

OVERVIEW

This study demonstrated that Activin A promotes osteoblastic differentiation in pre-osteoblasts Saos2 cells.

The main concern is that it is not clear what is the new findings/Innovative points of this study compared to previous studies. Previous studies (that are not concordant with the author’s conclusion) are not discussed. Some data are not enough to support the author’s findings. Further clarifications and more convincing evidence are required to support the authors’ conclusions.

Comments

Previous reports demonstrated that the ALK1-mediated pathway induced osteoblastic differentiation (29-32 references). What is the new finding of this study compared to previous studies?

What is the difference between this study and the study published by this group (Sugii et al. 2014, Bone)?

M&M and Abstract: Describe the origin of Saos2 and NOS1 cells. Human, mouse or rat?

Fig.3 and 5: No bar graph of controls.

Fig.1 (A), Fig.4 (A) and Fig.5 (B) : Osteoblastic layer is not clear. To show if osteoblasts express Activin A or ALK1, staining with osteoblast marker is required.

Fig.6: Control Bar graphs are hard to see.

Fig.6A: (siALK1+Ca+ActivinA) shows a decrease of total Smad1/5/9. How do you interpret this?

Fig.6B: There is no significace indication (P value) between (siControl+Ca+ActivinA) and (siALK1+Ca+ActivinA)

Discussion:

Previous author’s study (Sugii et al. 2014, Bone) showed that “Activin A down-regulating osteoblastic differentiation of HPDLCs”. This is not discussed.

Also, there are previous studies that demonstrated an inhibitory effect of activin A on osteoblastic differentiation. This is not discussed. 

The differences between the current study and the previous studies on the effects of Activin A in human osteosarcoma cell lines, (including SaOs-2) are not discussed.

Author Response

Thank you very much for your suggestion.

Q: Previous reports demonstrated that the ALK1-mediated pathway induced osteoblastic differentiation (29-32 references). What is the new finding of this study compared to previous studies?

A: Thank you for your question. Previous reports demonstrated that the ALK1-mediated pathway induced osteoblastic differentiation in myoblasts and mesenchymal stem cells (29-32 references), not in human pre-osteoblasts. In addition, BMP9 signaling was important for the ALK1-mediated osteoblastic differentiation in these cells. The new finding of this study is that Activin A could activate the ALK1-mediated osteoblastic differentiation in human pre-osteoblasts. We added this information in the study (p2, line62 – 64 in Introduction and p8, line240 – p9, line242 in Discussion).

Q: What is the difference between this study and the study published by this group (Sugii et al. 2014, Bone)?

A: I really appreciate your question. Periodontal tissue includes periodontal ligament (PDL) tissue, and alveolar bone. Once periodontal tissue has been irreversibly damaged by trauma, severe periodontitis, or deep caries, it is difficult to regenerate these tissues. Thus, we analyzed potential methods for enhancing the regeneration of periodontal tissue including PDL tissue and alveolar bone using Activin A. We previously focused on the effects of Activin A on human PDL cells (HPDLCs) using in vitro study and rat PDL tissue using in vivo study. We reported that Activin A promoted the proliferation, migration, and PDL-related gene expression of HPDLCs, and was expressed in PDL tissue and its expression was increased around damaged periodontal tissue along with expression of interleukin-1 (IL-1), which was co-expressed during the healing process in rat. We also observed Activin A expression on the surface of alveolar bone around damaged periodontal tissue. However, the biological functions of Activin A in pre-osteoblasts, which are distributed in alveolar bone, have not been reported. Thus, this study focused on the effects of Activin A on human pre-osteoblasts. We added this information in the study (p2, line66 – 75 in Introduction).

Q: Describe the origin of Saos2 and NOS1 cells. Human, mouse or rat?

A:Thank you for your question. The origin of Saos2 and NOS1 cells is human. We added the information of these cells in the study. (p10, line323 - 324 in Materials and Methods)

Q: Fig.3 and 5: No bar graph of controls. Fig.6: Control Bar graphs are hard to see.

A: I really appreciate your comment. We changed all figures to the higher resolution.

Q: Fig.1 (A), Fig.4 (A) and Fig.5 (B) : Osteoblastic layer is not clear. To show if osteoblasts express Activin A or ALK1, staining with osteoblast marker is required.

A: Thank you very much for your suggestion. We performed immunofluorescent analysis using an anti-Activin A antibody and an anti-Osteocalcin (OCN) antibody in rat periodontal tissue. Expression of Activin A and OCN was co-localized in osteoblastic layer of the alveolar bone. We exhibited the results in Fig. 1C of the study (p2, line87 – 89, p3, line90, 96 - 100 in Results).

Q: Fig.6A: (siALK1+Ca+ActivinA) shows a decrease of total Smad1/5/9. How do you interpret this?

A: Thank you for your question. As you mentioned, expression of total Smad1/5/9 had the tendency to decrease in siALK1+Ca+ActivinA group. However, when we quantified the expression of total Smad1/5/9, there had no significant differences between siALK1+Ca+ActivinA group and other groups (Fig.6C). In addition, expression levels of phospho-Smad1/5/9 were normalized against total-Smad1/5/9 expression and Activin A-induced phosphorylation of Smad1/5/9 was inhibited by siALK1 compared with control siRNA. Thus, we suggest that Activin A promotes osteoblastic differentiation of Saos2 cells through the ALK1-Smad1/5/9 pathway. We added this information in the study (p8, line221 – 223, 227, 232, and 233 in Results).

Q: Fig.6B: There is no significance indication (P value) between (siControl+Ca+ActivinA) and (siALK1+Ca+ActivinA)

A: Thank you very much for your suggestion. We forgot to add the significant difference between (siControl + Ca + Activin A) and (siALK1 + Ca + Activin A). We re-analyzed the values and added the significant difference (p8, line227, Fig. 6B).

Q: Previous author’s study (Sugii et al. 2014, Bone) showed that “Activin A down-regulating osteoblastic differentiation of HPDLCs”. This is not discussed. Also, there are previous studies that demonstrated an inhibitory effect of activin A on osteoblastic differentiation. This is not discussed.

A: Thank you very much for your suggestion. We showed that Activin A promoted osteoblastic differentiation of human pre-osteoblasts. On the other hand, as you mentioned, several studies have revealed that Activin A inhibited the osteoblastic differentiation of rat calvarial cells (1, 2) and we previously reported that Activin A inhibited osteoblastic differentiation of HPDLCs. These opposing effects of Activin A on osteoblastic differentiation may be caused by the basis of experimental conditions, such as cell type and differentiation status (3-5). Thus, we suggested that further studies elucidating the biological mechanism of Activin A-induced osteoblastic differentiation are required in the future study. We added this information in the study (p9, line254 – 261 in Discussion).

(1) Kawabata, N.; Kamiya, N.; Suzuki, N.; Matsumoto, M.; Takagi, M. Changes in

extracellular activin A:follistatin ratio during differentiation of a mesenchymal progenitor

cell line, ROB-C26 into osteoblasts and adipocytes. Life Sci 2007, 81, 8-18,

doi:10.1016/j.lfs.2007.04.011.

(2) Ikenoue, T.; Jingushi, S.; Urabe, K.; Okazaki, K.; Iwamoto, Y. Inhibitory effects of

activin-A on osteoblast differentiation during cultures of fetal rat calvarial cells. J Cell

Biochem 1999, 75, 206-214, doi:10.1002/(sici)1097-4644(19991101)75:2<206::aid-

jcb3>3.3.co;2-k.

(3) Bloise, E.; Ciarmela, P.; Dela Cruz, C.; Luisi, S.; Petraglia, F.; Reis, F.M. Activin A in

Mammalian Physiology. Physiol Rev 2019, 99, 739-780,

doi:10.1152/physrev.00002.2018.

(4) Nicks, K.M.; Perrien, D.S.; Akel, N.S.; Suva, L.J.; Gaddy, D. Regulation of

osteoblastogenesis and osteoclastogenesis by the other reproductive hormones, Activin

and Inhibin. Mol Cell Endocrinol 2009, 310, 11-20, doi:10.1016/j.mce.2009.07.001.

(5) de Jong, D.S.; van Zoelen, E.J.; Bauerschmidt, S.; Olijve, W.; Steegenga, W.T.

Microarray analysis of bone morphogenetic protein, transforming growth factor beta, and

activin early response genes during osteoblastic cell differentiation. J Bone Miner Res

2002, 17, 2119-2129, doi:10.1359/jbmr.2002.17.12.2119.

Q: The differences between the current study and the previous studies on the effects of Activin A in human osteosarcoma cell lines, (including SaOs-2) are not discussed.

A: Thank you very much for your suggestion. Previous study revealed that Activin A promoted the proliferation, invasion, and migration of human osteosarcoma cells (1). However, there was no reports about the effects of Activin A on the differentiation of human osteosarcoma cells. Our results showed for the first time that Activin A promoted the osteoblastic differentiation of human osteosarcoma cells, Saos2 cells and NOS1 cells. We added this information in the study (p9, line262 – 267 in Discussion).

(1) Zhu, J.; Liu, F.; Wu, Q.; Liu, X. Activin A regulates proliferation, invasion and migration in osteosarcoma cells. Mol Med Rep 2015, 11, 4501-4507, doi:10.3892/mmr.2015.3284.

Reviewer 2 Report

The aim of this paper is to determine the biological function of Activin A in alveolar bone and osteoblast differentiation due to the previous report showing Activin A, which was upregulated in damaged, inflamed periodontal tissue such as periodontal ligament (PDL) tissue. The main contribution of this paper shows that Activin A promotes osteoblastic differentiation through ALK-SMAD1/5/9 pathway. The strength of this paper is the clear, methodological way to determine the pathway involved in the osteoblastic differentiation. There was rigorous work with multiple methods to determine the ALK involved in pre-osteoblast to osteoblast differentiation by Activin A treatment. However, the connection between Activin A promoting healing of the alveolar bone and PDL tissue lacks rigor.

Due to the previous study, it is known that Activin A is increased in the damaged area. The damaged area is also known to have increased ILb and TNFa and was shown to increase ALK1. To make a stronger connection between the occurrence of Activin A increased in the damage area and ILb and TNFa increasing ALK1, treatment of the cells with Activin A compared to ILb and TNFa would be beneficial.

In addition, the proliferation assay that was performed in Figure 3B involves the mitochondria function. A proliferation assay timepoint for a longer period to match the other studies would be beneficial. Also, previous reports state Activin A can decrease calcium from inhibition of extracellular matrix (ECM) mineralization; what are the effects of Activin A without addition of calcium? It might be beneficial to test the ALP activity as well.

This article showed significant findings for the field and strong results to demonstrate the biological mechanism.

Author Response

Thank you very much for your suggestion.

Q: Due to the previous study, it is known that Activin A is increased in the damaged area. The damaged area is also known to have increased IL1b and TNFa and was shown to increase ALK1. To make a stronger connection between the occurrence of Activin A increased in the damage area and IL1b and TNFa increasing ALK1, treatment of the cells with Activin A compared to IL1b and TNFa would be beneficial.

A: Thank you very much for your suggestion. To analyze the connection between the occurrence of Activin A increased in the damage area and IL-1 and TNF- increasing ALK1, we analyzed the expression of ALK1 in Activin A-treated Saos2 cells and we found that expression of ALK1 was upregulated by Activin A treatment in Saos2 cells. In addition, expression of ALK1 was further upregulated in Activin A-treated Saos2 cells after IL-1 or TNF- treatment (Supplemental Fig.8). These results suggest that Activin A acts on the autocrine or paracrine manner in pre-osteoblasts and Activin A and ALK1 play an important role in the healing of periodontal tissue including alveolar bone, especially during the inflammatory process. We added the results and discussion in the study (p6, line193 – 197 in Results and p9, line294 – p10, line303 in Discussion).

Q: The proliferation assay that was performed in Figure 3B involves the mitochondria function. A proliferation assay timepoint for a longer period to match the other studies would be beneficial.

A: I really appreciate your comment. We analyzed the proliferation assay at day 5 and day7. There were no significant differences between Control, siControl, and siALK1 groups in Saos2 cells. We exhibited the results in Fig. 3B of the study (p5, line150 in Results).

Q: Previous reports state Activin A can decrease calcium from inhibition of extracellular matrix (ECM) mineralization; what are the effects of Activin A without addition of calcium? It might be beneficial to test the ALP activity as well.

A: Thank you very much for your suggestion. To analyze the effects of Activin A without addition of calcium, we analyzed ALP activity and gene expression of ALP in Saos2 cells treated with or without CaCl2 and Activin A. ALP activity and ALP expression were upregulated in Activin A-treated Saos2 cells with CaCl2 compared with Saos2 cells with CaCl2. On the other hand, Saos2 cells without CaCl2 exhibited no promotion effects of ALP activity and ALP expression, and Activin A-treated Saos2 cells without CaCl2 also showed no effects on ALP activity and ALP expression

(Supplemental Fig. 1). The past report showed that Activin A increased the intracellular Ca2+ concentration (1). These results suggest that Activin A promoted osteoblastic differentiation of Saos2 cells through increasing the intracellular Ca2+ concentration. However, as you mentioned, previous study has revealed that Activin A inhibited ALP activity by inhibition of extracellular matrix mineralization in human bone marrow derived mesenchymal stem cells (2). These opposing effects of Activin A on ALP activity may be caused by the basis of experimental conditions, such as cell type and differentiation status (3-5). We added the results and discussion in the study (p4, line108 – 113 in Results and p9, line250 – 261 in Discussion).

(1) Shibata, H.; Yasuda, H.; Sekine, N.; Mine, T.; Totsuka, Y.; Kojima, I. Activin A

increases intracellular free calcium concentrations in rat pancreatic islets. FEBS Lett 1993,

329, 194-198, doi:10.1016/0014-5793(93)80220-o.

(2) Alves, R.D.; Eijken, M.; Bezstarosti, K.; Demmers, J.A.; van Leeuwen, J.P. Activin A

suppresses osteoblast mineralization capacity by altering extracellular matrix (ECM)

composition and impairing matrix vesicle (MV) production. Mol Cell Proteomics 2013,

12, 2890-2900, doi:10.1074/mcp.M112.024927.

(3) Bloise, E.; Ciarmela, P.; Dela Cruz, C.; Luisi, S.; Petraglia, F.; Reis, F.M. Activin A in

Mammalian Physiology. Physiol Rev 2019, 99, 739-780,

doi:10.1152/physrev.00002.2018.

(4) Nicks, K.M.; Perrien, D.S.; Akel, N.S.; Suva, L.J.; Gaddy, D. Regulation of

osteoblastogenesis and osteoclastogenesis by the other reproductive hormones, Activin

and Inhibin. Mol Cell Endocrinol 2009, 310, 11-20, doi:10.1016/j.mce.2009.07.001.

(5) de Jong, D.S.; van Zoelen, E.J.; Bauerschmidt, S.; Olijve, W.; Steegenga, W.T.

Microarray analysis of bone morphogenetic protein, transforming growth factor beta, and

activin early response genes during osteoblastic cell differentiation. J Bone Miner Res

2002, 17, 2119-2129, doi:10.1359/jbmr.2002.17.12.2119.

Round 2

Reviewer 1 Report

The revised manuscript is considerably improved in its present form and it is suitable for publication in IJMS.

Reviewer 2 Report

As previously stated, the aim of this paper is to study Activin A biological function  and osteoblast differentiation in alveolar bone. Previous report showed Activin A was upregulated in damaged, inflamed periodontal tissue such as periodontal ligament (PDL) tissue which led to the researchers studies here. The main scientific contribution is Activin A promotes osteoblastic differentiation through ALK-SMAD1/5/9 pathway. The interesting results added based edits with opposing effects of Activin A on ALP activity from previous study demonstrate the importance of studying Activin A and ALP in specific cell types and different stages of differentiation. 

This article showed significant findings for the field and strong results to demonstrate the biological mechanism.